# XTSFormer: Cross-Temporal-Scale Transformer for Irregular Time Event Prediction

## Abstract

Event prediction aims to forecast the time and type of a future event based on a historical event sequence. Despite its significance, several challenges exist, including the irregularity of time intervals between events, cycles, periodicity, and the complex multi-scale nature of event interactions, as well as the potentially high computational costs for long event sequences. However, current neural temporal point processes (TPPs) methods do not capture the multi-scale nature of event interactions, which is common in many real-world applications such as clinical event data. To address these issues, we propose the cross-temporal-scale transformer (XTSFormer), designed specifically for irregularly timed event data. Our model comprises two vital components: a novel Feature-based Cycle-aware Positional Encoding (FCPE) that adeptly captures the cyclical nature of time, and a hierarchical multi-scale temporal attention mechanism. These scales are determined by a bottom-up clustering algorithm. Extensive experiments on several real-world datasets show that our XTSFormer outperforms several baseline methods in prediction performance.

## 1 Introduction

Given a sequence of historical events with timestamps and types, event prediction aims to predict the time and type of the next event. This problem is common in many application domains, such as user behavior modeling in recommender systems (Wang et al., 2021; Yan et al., 2019), fraud detection in financial transactions (Bacry et al., 2015), and disease modeling based on electronic health records (EHRs) (Tomašev et al., 2021; Duan et al., 2019; Liu et al., 2022b). In the health domain, adverse events related to unsafe care are among the top ten causes of death in the U.S. (Dingley et al., 2011; Weinger et al., 2003). The large volume of EHR data being collected in hospitals, together with recent advancements in machine learning and artificial intelligence, provide unique opportunities for data-driven and evidence-based clinical decision-making systems. Event prediction by learning common clinical operation patterns can identify anomalous or potentially erroneous clinical operations deviating from clinical practice guidelines before they occur, e.g., forgetting a certain medication or lab order by mistake. The learned event patterns can also facilitate generating real-world evidence through retrospective analysis of EHRs to help design improvement of clinical operations in medical practice. Figure 1 presents an illustrative example, where the event sequence represents a patient's medication intake. In this sequence, medication type 1 is taken nearly every 12 hours, and medication type 2 is taken about every two days. This scenario exhibits multi-scale and cyclic patterns commonly observed in healthcare event data.

However, this domain presents several challenges. First, the irregularity of time intervals between events makes common time series prediction methods insufficient. Second, the event sequence patterns exhibit cycles, periodicity, and the multi-scale effect. For example, clinical operational events such as medication administration in operating rooms occur on a fine scale, typically within minutes. Conversely, events pre- or post-operation are on a coarser scale, often spanning hours or days. Similarly, in the realm of consumer behavior, purchasing frequency markedly rises during festivals compared to normal days, which see less frequent buying activities. Both scenarios exemplify the multi-scale nature of events in different contexts. However, accurately modeling these complex patterns, especially within extended event sequences, can involve high computational costs.

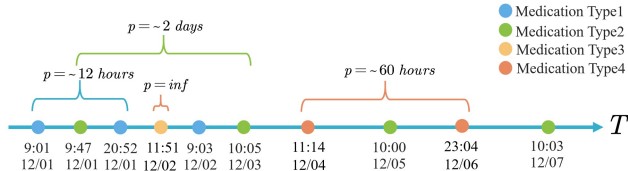

Figure 1: An example of medication taken sequence in EHR.

Existing methods are generally based on the temporal point processes (TPPs), a common framework for modeling asynchronous event sequences in continuous time (Cox & Isham, 1980; Schoenberg et al., 2002). Traditional statistical TPPs models (Daley & Vere-Jones, 2008) characterize the stochastic nature of event timing but can only capture simple patterns in event occurrences such as self-excitation (Hawkes, 1971). More recently, deep learning methods (also called neural TPPs) have become more popular due to the capability of modeling complex event dependencies in the intensity function of TPPs. Recurrent neural networks (RNNs), known for their inherent ability to process sequential data, were first introduced to neural TPPs, with models like Recurrent Marked Temporal Point Process (RMTPP) (Du et al., 2016), continuous-time LSTM (CT-LSTM) (Mei & Eisner, 2017), and Intensity Function-based models (Xiao et al., 2017; Omi et al., 2019). While LSTM-based approaches address some challenges like vanishing gradients, other issues such as long-range dependencies remained unresolved. Transformers (Vaswani et al., 2017), leveraging their self-attention mechanism, circumvent the long-range dependency problem by allowing direct interactions between all events in a sequence. For example, Transformer Hawkes Process (THP) (Zuo et al., 2020) leverages the self-attention mechanism to efficiently capture long-term term dependencies across varied event sequence data. Meanwhile, Self-Attentive Hawkes Process (SAHP) (Zhang et al., 2020) employs the translation of time intervals into phase shifts of sinusoidal functions, coupled with self-attention, as a strategy for enhanced feature learning. Moreover, (Yang et al., 2022) enhance the neural Hawkes process (Mei & Eisner, 2017) by replacing its architecture with a flatter attention-based model. However, these methods do not capture the important multi-scale patterns within event sequences. There are some works on multi-scale transformers (e.g., Scaleformer (Shabani et al., 2023) and Pyraformer (Liu et al., 2022a)) and efficient transformers (e.g., LogTrans (Li et al., 2019) and Informer (Zhou et al., 2021)) for time series data, but these methods assume regular time intervals and thus are not applicable for irregular time event prediction.

To address these challenges, we propose a novel cross-temporal-scale transformer (XTSFormer) for irregular time event prediction. Our XTSFormer consists of Feature-based Cycle-aware positional encoding (FCPE) and cross-scale temporal attention within a multi-scale time hierarchy. Specifically, we define the time scale on irregular time event sequences by the merging order of a bottom-up clustering algorithm (e.g., agglomerative). The intuition is that events with shorter intervals (at smaller scales) will be merged earlier. of irregular time events. We designed a cross-scale attention operation by specifying the key set as nodes in the same scale level. In summary, we make the following contributions.

- Our XTSFormer is the first work to take into account multi-scale features in neural TPPs. Such multi-scale patterns are of important practical value in clinical event analysis.

- We introduce a novel time positional encoding, the Feature-based Cycle-aware Positional Encoding (FCPE), which incorporates both feature and cyclical information. This approach strengthens our model's capacity to capture complex temporal patterns.

- We design a cross-scale attention operation within the multi-scale hierarchy and compare the time cost with default all-pair attention.

- Experiments on two public datasets and two real patient safety datasets demonstrate the superior performance of our proposed XTSformer. We present a detailed evaluation that shows our model outperforms established benchmarks, thereby validating the effectiveness of our methodological contributions.

## 2 PROBLEM STATEMENT

Consider a temporal sequence $\mathcal{Q}$ of events denoted as $< e_1, ..., e_i, ..., e_L >$, where $L$ represents the sequence length. Each event, $e_i$, can be characterized by a pair $(t_i, k_i)$: $t_i$ signifies the event time, and $k_i \in \{1, 2, ..., n_K\}$ indicates the event type, with $K$ denoting the total number of type classes. The objective of the event prediction problem is to predict the subsequent event, $e_{L+1} = (t_{L+1}, k_{L+1})$. It's important to note that the time of each event, $t_i$, is irregular, i.e., events don't occur at fixed intervals. These event times can exhibit patterns across various temporal scales. To illustrate, clinical operational events like medication administration may be recorded at minute intervals within an operation room, yet they might be logged hourly when noted pre- or post-operation.

## 3 METHODOLOGY

### 3.1 OVERALL MODEL ARCHITECTURE

This section introduces our proposed cross-temporal-scale transformer (XTSFormer) model, as illustrated in Figure 2. Our main idea is to establish a multi-scale time hierarchy and conduct cross-scale attention with selective key sets in each scale. Latent features are processed using pooling operations across multiple scale levels. In order to capture complex cyclic patterns in irregular time, we design a Feature-based Cycle-ware time Positional Encoding (FCPE).

Starting from the irregular time event sequence, we first do a bottom-up clustering to specify multiple-scale hierarchy of event points. This is done in the preprcessing phase. Within our model, we start with embedding operations, including our FCPE and semantic feature embedding as shown in Figure 2(a). Then we do cross-scale attention operations, from the smallest scale till the largest scale as shown in Figure 2(b). For each scale, our model conducts cross-scale attention at the same scale level, an average pooling operation (based on the tree hierarchy), and concatenation of pooled clusters with the clusters in the next scale. The iterations continue until reaching the root node. This process learns complex multi-scale representation within a multi-level hierarchy without sacrificing granularity or specificity. Moreover, the approach enhances computational efficiency by reducing the size of a key set in cross-attention operations. Finally, our model incorporates a decoder that leverages the representation of the entire past event sequence. This decoder is adept at forecasting both the type and timing of future events, utilizing a Weibull distribution for accurate predictions, as depicted in Figure 2(c).

### 3.2 FEATURE-BASED CYCLE-AWARE TIME POSITIONAL ENCODING

The goal of temporal position encoding is to capture the relative timespan between events. Previous temporal positional encoding methods have typically focused on encoding a continuous timestamp only. However, it has been shown that semantic features can play a crucial role in accurately capturing the periodic patterns in many real-world phenomena (Ke et al., 2021; Zhang et al., 2019). Thus, we introduce a novel feature-based cycle-aware time positional encoding (FCPE).

Time positional encoding can be described as a function $\mathcal{P} : T \to \mathbb{R}^{d \times 1}$, mapping from the time domain $T \subset \mathcal{R}$ to a $d$-dimensional vector space. In attention mechanisms, it is the dot product of time positional encodings that matters (Xu et al., 2019). Therefore, the relative timespan $|t_a - t_b|$ between events $a$ and $b$ implies crucial temporal information. Considering events $a$ and $b$, we define a temporal kernel $\mathcal{K} : T \times T \to \mathbb{R}$, such that

$$\mathcal{K}(t_a, t_b) = \mathcal{P}(t_a) \cdot \mathcal{P}(t_b) = \mathcal{F}(t_a - t_b), \tag{1}$$

where $\mathcal{F}$ is a location invariant function of the timespan. As proofed in Appendix A.1.1, the kernel $\mathcal{K}$ defined above satisfies the assumptions of Bochner's Theorem, as stated below:

**Theorem 1** *(Bochner's Theorem). A continuous, translation-invariant kernel $\mathcal{K}(t_a, t_b) = \mathcal{F}(t_a - t_b)$ is positive definite if and only if there exists a non-negative measure on $\mathbb{R}$ such that $\mathcal{F}$ is the Fourier transform of the measure.*

Given this, the kernel $\mathcal{K}$ can be represented as Eq. (2):

$$\mathcal{F}(t_a - t_b) = \int_{-\infty}^{\infty} e^{iw(t_a - t_b)} p(w) \, dw. \tag{2}$$

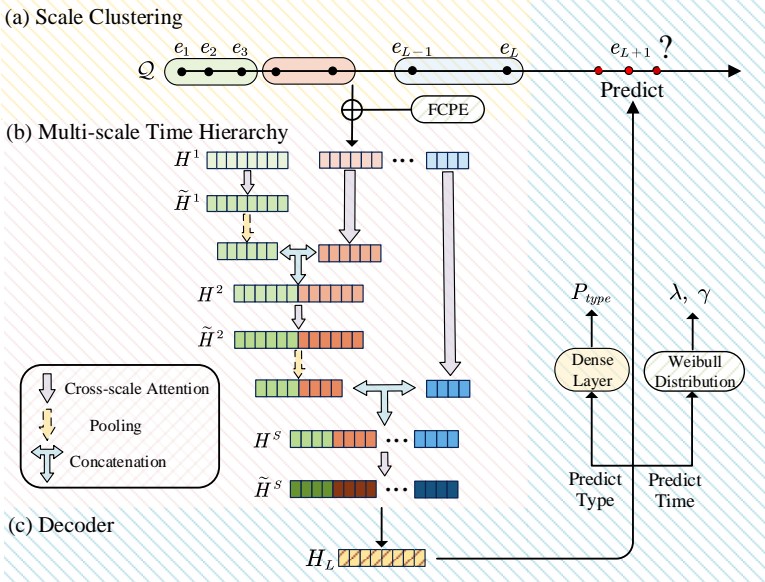

Figure 2: The flowchart of the proposed XTSFormer, which consists of three parts: (a) scale clustering; (b) multi-scale time hierarchy; (c) event time and type prediction (decoder).

Different from (Xu et al., 2020), which uses the Monte Carlo integral to approximate the expectation of $\mathcal{F}$, we sample the probability density $p(w_k)$ on several frequencies $w_k$ and learn $p(w_k)$ based on the event feature, where $k = 0, ..., \frac{d}{2} - 1$ ($d$ is an even integer). The frequencies $w_k$ are learnable parameters initialized as $\frac{2\pi k}{\frac{d}{2}}$, corresponding to the Discrete Fourier Transform (DFT) of the spectral density function, as follows.

$$\mathcal{F}(t_a - t_b) \approx \sum_{k=1}^{\frac{d}{2}} \mu(k) e^{i2\pi k \frac{t_a - t_b}{\frac{d}{2}}} = \sum_{k=1}^{\frac{d}{2}} \mu(k) \cos\left(2\pi k \frac{t_a - t_b}{\frac{d}{2}}\right), \quad (3)$$

where $\mu(k)$ (representing $p(w_k)$) is the non-negative power spectrum at the frequency index $k$ and $\frac{d}{2}$ denotes the number of frequencies. Considering that $w_k$ is learnable, we have

$$\mathcal{F}(t_a - t_b) = \sum_{k=1}^{\frac{d}{2}} \mu^k \cos(w_k(t_a - t_b)), \quad (4)$$

where $\mu^k$ is the learned probability density corresponding to frequency $w_k$.

Thus in accordance with the above conditions and to satisfy Eq. (1) and Eq. (4), we propose the final positional encoding $\mathcal{P}(t_i)$ for time $t_i$ as shown in Eq. (5), where $d$ is the encoding dimension, $w_k$ is the $k$-th sample of frequency, and $\mu_i^k$ is the learned feature-based probability density corresponding to $w_k$. Specifically, $\mu_i = [\mu_i^1, \mu_i^2, ..., \mu_i^{\frac{d}{2}}]^T$ can be expressed as $\mu_i = W^\mu \mathbf{k_i}$, where $W^\mu \in \mathbb{R}^{\frac{d}{2} \times K}$ is a learnable parameter matrix, $\mathbf{k_i} \in \mathbb{R}^{K \times 1}$ is the one-hot encoding of event type $k_i$.

$$\mathcal{P}(t_i) = [\mu_i^1 \cos(w_1 t_i), \mu_i^1 \sin(w_1 t_i), ..., \mu_i^{\frac{d}{2}} \cos\left(w_{\frac{d}{2}} t_i\right), \mu_i^{\frac{d}{2}} \sin\left(w_{\frac{d}{2}} t_i\right)]^T \in \mathbb{R}^{d \times 1}. \quad (5)$$

The advantages of our FCPE are twofold. First, it builds upon the assumption that any point in time can be represented as a vector derived from a series of sine and cosine functions that reflect the cyclical nature of time with varying intensities and frequencies, which is suitable for modeling irregular time interval. Second, we propose learning the intensities associated with each sample frequency

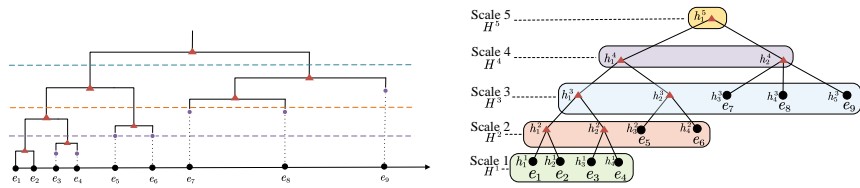

(a) Agglomerative clustering of irregular time (b) Scale hierarchy based on merging order.
points.

Figure 3: An illustration of multi-scale hierarchy on irregular time points by bottom-up clustering.

based on the event semantic features (e.g., event type) at a particular time. Ideally, event types that occur frequently will be reflected in higher density values $\mu^k$ on the higher frequency $w_k$. As shown in Appendix A.1.2, FCPE's translation invariance ensures stability, sustaining performance even when there are shifts in the input feature.

After temporal position encoding with FCPE, we add it into a non-temporal feature embedding, i.e., $f_i = W^k \mathbf{k}_i + \mathcal{P}_i$, where $f_i$ is the entire embedding $i$-th event, $W^k \in \mathbb{R}^{d \times K}$ are learnable parameters for non-temporal embedding, $\mathbf{k_i}$ is the one-hot encoding of event type $k_i$, and $\mathcal{P}_i \in \mathbb{R}^{d \times 1}$ is the FCPE of time $t_i$.

### 3.3 CROSS-SCALE ATTENTION ON IRREGULAR-TIME SEQUENCE

One unique challenge of designing cross-scale attention on irregular time event sequences is that there is no obvious definition of temporal scales. This is different from regular time series data, where different temporal scales can be easily defined based on the original or downsampled resolutions. For irregular time event sequences, intuitively, events that occur with short time intervals tend to interact with each other at a small time scale (e.g., medications every few minutes within the operation room), while events with longer intervals are at a large time scale (e.g., medication every few days post-operation). Here we would like to establish the temporal scale concept on irregular time sequences in an algorithmic manner. We do this by hierarchical clustering.

#### 3.3.1 MULTI-SCALE TIME HIERARCHY

We define the temporal scales on irregular time points through a bottom-up hierarchical clustering, e.g., agglomerative (Tan et al., 2016; Day & Edelsbrunner, 1984). The agglomerative algorithm starts with each time point as an initial cluster and recursively merges the two closest clusters each time (measured by the minimum, maximum, or centroid distance among point pairs), until all intermediate clusters are merged into one. The greedy criteria used in the agglomeration algorithm enforce that time points that are closer to each other (in smaller scales) be merged earlier. Therefore, we can determine the temporal scale based on the cluster merging order in a multi-level hierarchy. Figure 3 illustrates this process with an example. There are nine events $e_1$ to $e_9$. Figure 3(a) shows the bottom-up clustering process with one merging operation at a time. The merging order of intermediate clusters is shown by the levels of vertical bars. In this example, points $e_1$ to $e_2$ are merged first, then $e_3$ and $e_4$, followed by $e_5$ and $e_6$. Next, two leftmost intermediate clusters are merged. The process continues until all clusters are merged into a root node. The merging order indicates that $e_7$ and $e_8$ are at a larger time scale than $e_1$ and $e_2$. This is consistent with our intuition when looking at the point distribution.

To quantify the time scales of event points, we can vertically slice the merging order of all initial and intermediate clusters into different intervals. Those clusters that are merged in the $s$-th vertical interval from the bottom are in the scale of $s$. For example, in Figure 3(a), we can use three thresholds to split merging operations into four intervals. Within each interval, we can check the initial clusters before any merging in the interval and the final clusters before this interval ends. For example, $e_1$ and $e_2$ as well as $e_3$ and $e_4$ are merged into two internal nodes (red triangles) in the first (bottom) interval. Therefore, they are in scale 1. As another instance of example, $e_7$, $e_8$, and $e_9$ are merged in the third interval, and thus they are in scale 3. The cumulative merging process can be summarized

in a hierarchical tree structure like Figure 3(b), in which the **temporal scale** of a tree node can be defined by its level.

One important question is how to choose the slicing thresholds. The thresholds control the granularity of multiple scales. If we need fine-grained multi-scale levels, we need to set up a larger number of thresholds (intervals, or tree levels). In an extreme case, the total number of levels equals the number of event points. In practice, such a scale choice is likely inefficient. To control the number of (leaf or internal) points at each scale level, we can configure the slicing thresholds based on the number of merging operations. The multi-scale hierarchy in Figure 3(b) can be configured by setting the number of merging operations as 2, 2, 3, and 1 respectively. This will help us manage the number of intermediate clusters at each scale.

We denote the latent representations at different tree nodes in each scale level $s$ as $H^s = [h_1^s, h_2^s, ..., h_{n_s}^s]$, where $h_j^s$ is the $j$-th node, and $n_s$ is the number of nodes in scale $s$. Note that $h_j^s = f_j$ (raw embedding) for a leaf node ($e_j$) at the beginning. In Figure 3(b), there are four node representations in the 1st scale, four in the 2nd scale, and so on.

### 3.3.2 CROSS-SCALE ATTENTION

We now introduce our attention operation in the multi-scale time hierarchy. In the common all-pair attention, for each time point (query), we have to compute its attention weights to all points (keys). In our cross-scale temporal attention, for each tree node (query), we only do temporal attention on a **selective key set**, i.e., nodes in the same scale level. The cross-attention operation is expressed in Eq. (6), where $\tilde{h_j^s}$ is the representation of $h_j^s$ after cross-attention, $\boldsymbol{q}_j^s$ is the query vector for $j$th node at scale $s$, $\boldsymbol{k}_l^s$ is the key vector for the $l$-th node, $\boldsymbol{v}_l$ is the value vector, $\mathcal{N}_j^s$ as the selective key set of $h_j^s$, and $D_K$ is the dimension of key and query vectors as a normalizing term. Consider the example in Figure 3(b). The key set for $e_6$ ($h_4^2$) has four nodes ($h_1^2$, $h_2^2$, $h_3^2$, and $h_4^2$), including itself. This reduces the total number of keys from 9 to 4.

$$\tilde{h_j^s} = \sum_{l \in \mathcal{N}_j^s} \frac{\exp(\boldsymbol{q}_j^s \boldsymbol{k}_l^{sT}/\sqrt{D_K})\boldsymbol{v}_l}{\sum_{l \in \mathcal{N}_j^s} \exp(\boldsymbol{q}_j^s \boldsymbol{k}_l^{sT}/\sqrt{D_K})}. \tag{6}$$

### 3.3.3 TIME COST ANALYSIS

The main advantages of the cross-scale transformer include that it captures the multi-scale hierarchical patterns and more importantly that it can reduce the attention operations. Assume the number of input temporal points is $\hat{L}$, batch size is $B$, number of heads is $h$, and hidden dimension is $d$. In our attention computation, the query matrix dimensions is represented by $Q \in \mathbb{R}^{B \times h \times L \times d}$. For each query point $\mathcal{Q}_i$ (where $1 \leq i \leq \hat{L}$), its selective key set size is $M$ (depend on the threshold in each level). Thus the key matrix results from concatenating key sets of all query points, yielding a dimension of $\mathbb{R}^{B \times h \times \hat{L} \times M \times d}$. Consequently, the Flops of the attention computation with our approach is $B \cdot h \cdot \hat{L} \cdot M \cdot d$, with the key set size $M$ scaling $O(log\hat{L})$. This results in a significantly more efficient attention computation Flops, specifically $B \cdot h \cdot \hat{L} \cdot \log \hat{L} \cdot d$, compared to the vanilla transformer computation of $B \cdot h \cdot \hat{L}^2 \cdot d$.

## 3.4 DECODER AND LOSS FUNCTION

Our decoder comprises two parts: predicting event type and event time, as shown in Figure 2(c). We first derive a comprehensive latent representation, $H_L$, of the entire past event sequence. This is achieved by applying a dense layer to $\tilde{H}^s$. Given that $\tilde{H}^s$ consolidates features across various scales, it effectively captures the temporal patterns of the upcoming item we aim to predict.

### 3.4.1 EVENT TYPE PREDICTION

To predict the type of the next event based on the latent embedding $H_L$ of past events, we utilize a dense transformation layer, which is then followed by a softmax function. This process yields the predicted probability distribution of the event type $P_{type,i}$. Given the true event type labels $y_{true}$, we compute the cross-entropy loss: $\mathcal{L}_p = -\sum_i y_{true,i} \log(P_{type,i})$.

Table 1: Results (average $\pm$ std) of all methods on Medications and Providers dataset.

| Methods | Medications | | | | Providers | | | |
|---|---|---|---|---|---|---|---|---|
| | Accuracy (%) | F1-score (%) | RMSE | NLL | Accuracy (%) | F1-score (%) | RMSE | NLL |
| HP | $21.9_{\pm 1.1}$ | $18.1_{\pm 2.1}$ | $2.78_{\pm 0.33}$ | $3.54_{\pm 0.38}$ | $32.1_{\pm 2.5}$ | $31.9_{\pm 2.6}$ | $5.17_{\pm 1.30}$ | $2.19_{\pm 0.13}$ |
| RMTPP | $23.4_{\pm 0.6}$ | $20.1_{\pm 1.8}$ | $1.87_{\pm 0.77}$ | $3.10_{\pm 0.18}$ | $35.7_{\pm 2.1}$ | $33.2_{\pm 2.7}$ | $4.11_{\pm 1.40}$ | $2.23_{\pm 0.11}$ |
| CTLSTM | $22.5_{\pm 0.6}$ | $19.2_{\pm 1.7}$ | $1.61_{\pm 0.41}$ | $3.23_{\pm 0.18}$ | $34.5_{\pm 1.4}$ | $32.5_{\pm 1.9}$ | $3.12_{\pm 1.50}$ | $1.93_{\pm 0.08}$ |
| SAHP | $28.4_{\pm 0.9}$ | $25.5_{\pm 2.1}$ | $1.81_{\pm 0.30}$ | $2.44_{\pm 0.21}$ | $38.0_{\pm 1.9}$ | $\mathbf{37.2}_{\pm 2.1}$ | $3.55_{\pm 1.93}$ | $2.10_{\pm 0.09}$ |
| THP | $27.1_{\pm 0.7}$ | $26.1_{\pm 1.3}$ | $1.41_{\pm 0.33}$ | $2.49_{\pm 0.19}$ | $37.5_{\pm 2.2}$ | $33.8_{\pm 1.9}$ | $2.84_{\pm 1.48}$ | $1.82_{\pm 0.09}$ |
| A-NHP | $30.2_{\pm 0.5}$ | $25.5_{\pm 0.8}$ | $1.57_{\pm 0.29}$ | $2.54_{\pm 0.22}$ | $38.9_{\pm 1.5}$ | $34.9_{\pm 1.5}$ | $2.89_{\pm 1.54}$ | $1.83_{\pm 0.11}$ |
| **XTSFormer** | $\mathbf{33.5}_{\pm 0.8}$ | $\mathbf{29.4}_{\pm 1.1}$ | $\mathbf{1.12}_{\pm 0.24}$ | $\mathbf{2.23}_{\pm 0.20}$ | $\mathbf{43.9}_{\pm 1.3}$ | $\mathbf{37.2}_{\pm 1.5}$ | $\mathbf{2.33}_{\pm 1.74}$ | $\mathbf{1.75}_{\pm 0.10}$ |

Table 2: Results (average $\pm$ std) of all methods on Financial and StackOverflow dataset.

| Methods | Financial | | | | StackOverflow | | | |
|---|---|---|---|---|---|---|---|---|
| | Accuracy (%) | F1-score (%) | RMSE | NLL | Accuracy (%) | F1-score (%) | RMSE | NLL |
| HP | $50.8_{\pm 1.8}$ | $48.7_{\pm 1.8}$ | $2.92_{\pm 0.87}$ | $3.24_{\pm 0.13}$ | $37.5_{\pm 1.3}$ | $28.1_{\pm 2.3}$ | $9.45_{\pm 2.23}$ | $3.87_{\pm 0.11}$ |
| RMTPP | $52.0_{\pm 1.9}$ | $52.9_{\pm 2.0}$ | $1.56_{\pm 0.58}$ | $3.60_{\pm 0.26}$ | $41.9_{\pm 3.5}$ | $28.9_{\pm 2.2}$ | $9.78_{\pm 1.85}$ | $3.76_{\pm 0.09}$ |
| CTLSTM | $68.8_{\pm 2.6}$ | $64.2_{\pm 2.5}$ | $1.89_{\pm 0.50}$ | $2.11_{\pm 0.41}$ | $44.5_{\pm 1.9}$ | $33.4_{\pm 1.8}$ | $5.86_{\pm 1.19}$ | $2.70_{\pm 0.15}$ |
| SAHP | $74.6_{\pm 0.8}$ | $75.2_{\pm 1.4}$ | $3.52_{\pm 0.41}$ | $1.30_{\pm 0.24}$ | $43.2_{\pm 1.8}$ | $\mathbf{35.1}_{\pm 2.1}$ | $5.57_{\pm 0.79}$ | $2.83_{\pm 0.06}$ |
| THP | $72.4_{\pm 2.1}$ | $73.5_{\pm 2.6}$ | $1.30_{\pm 0.33}$ | $1.33_{\pm 0.15}$ | $46.8_{\pm 2.3}$ | $34.6_{\pm 1.7}$ | $4.99_{\pm 1.18}$ | $2.85_{\pm 0.07}$ |
| A-NHP | $72.8_{\pm 1.2}$ | $72.4_{\pm 2.3}$ | $0.93_{\pm 0.47}$ | $\mathbf{1.16}_{\pm 0.27}$ | $46.8_{\pm 1.4}$ | $33.7_{\pm 2.2}$ | $5.85_{\pm 0.75}$ | $2.05_{\pm 0.05}$ |
| **XTSFormer** | $\mathbf{77.5}_{\pm 1.9}$ | $\mathbf{78.8}_{\pm 1.3}$ | $\mathbf{0.88}_{\pm 0.41}$ | $1.18_{\pm 0.11}$ | $\mathbf{49.4}_{\pm 1.1}$ | $35.0_{\pm 1.9}$ | $\mathbf{4.21}_{\pm 0.60}$ | $\mathbf{1.89}_{\pm 0.07}$ |

### 3.4.2 EVENT TIME PREDICTION

For event time prediction, we add another dense layer on top of $H_L$ to learn the distribution parameters, scale parameter $\lambda$ and shape parameter $\gamma$, of the temporal point process, e.g., intensity in the Poisson point process. Here we adopt the Weibull distribution (Rinne, 2008) to model the intensity function, in contrast to the commonly used exponential distribution. The exponential distribution is a special case of the Weibull distribution with $\gamma = 1$. Its constant intensity function implies events occur with consistent likelihood, independent of past occurrences. This lack of historical memory limits its suitability in contexts influenced by past events. In contrast, the Weibull distribution's hazard function can be increasing, decreasing, or constant, providing a versatile framework to model the influence of past events on future likelihoods. We evaluate the effectiveness of these two distributions as intensity functions in our subsequent experimental section.

We use the negative log likelihood (NLL) of event time as the loss function for event time prediction:

$$\mathcal{L}_t = -\log P\left(t'; \lambda, \gamma\right) = -\log\left(\frac{\gamma}{\lambda}\left(\frac{t'}{\lambda}\right)^{\gamma-1} e^{-\left(\frac{t'}{\lambda}\right)^{\gamma}}\right), \tag{7}$$

where $t'$ is the label time. The final loss is $\mathcal{L} = (1-\alpha)\mathcal{L}_t + \alpha\mathcal{L}_p$, where $\alpha$ is a hyperparameter for trade-off.

## 4 EXPERIMENTAL EVALUATION

The goal of the evaluation section is to compare our proposed XTSFormer with baseline models in neural TPPs in prediction accuracy (both event time and event type). We also want to understand the importance of different components within our model through an ablation study. For the event type prediction task, given a large number of classes, we utilized the accuracy and the F1-score as evaluation metrics. Meanwhile, for the event time prediction task, the root mean square error (RMSE) and NLL was chosen as the performance metric. The detailed experimental setup is in Appendix A.2.1.

### 4.1 DATASETS AND COMPARATIVE METHODS

In our experiments, four benchmark datasets were used, comprising two Electronic Health Record (EHR) datasets (Medications and Providers) from our university hospital, Financial Transactions (Du et al., 2016), and StackOverflow (Leskovec & Krevl, 2014), as summarized in Table 4 (details are in Appendix A.2.2).

The comparative methods include one traditional algorithm, *i.e.,* Hawkes Processes (HP) (Zhang et al., 2020), two RNN-based algorithms, *i.e.,* RMTPP (Du et al., 2016) and CT-LSTM (Mei & Eisner, 2017) , and three Transformer-based algorithms, *i.e.,* SAHP (Zhang et al., 2020), THP (Zuo et al., 2020), and A-NHP(Yang et al., 2022).

## 4.2 COMPARISON ON PREDICTION ACCURACY

Table 1 and Table 2 summarize the accuracy, F1-score, RMSE, and negative log likelihood (NLL) of all candidate methods on four real-world datasets. We can see that the traditional Hawkes process model (HP) has the lowest accuracy in event-type prediction. The RNN-based models perform slightly better than HP in overall accuracy and F1-score. The Transformer-based models are generally more accurate than the RNN-based models. Their overall accuracy is around 4% to 5% higher than RNN-based models. Among transformer models, XTSFormer performs the best, whose overall accuracy is 3% to 5% higher than other transformers. This could be explained by the fact that our model captures the multi-scale temporal interactions among events. For event time prediction, we observe similar trends, except that the RMSE of event time prediction for SAHP is somehow worse than other transformers (close to the RNN-based models). The reason could be that the event sequences in our real-world datasets do not contain self-exciting patterns as assumed in the Hawkes process.

## 4.3 ABLATION STUDY

To evaluate the effectiveness of our proposed model components, we conducted an ablation study on various datasets, including Medications, Providers, Financial, and StackOverflow. The study investigates the impact of Feature-based Cycle-Aware Positional Encoding (FCPE), multi-scale temporal attention, and choice of event time distribution (Exponential or Weibull). Table 3 shows the accuracy results of the ablation study. Specifically, we compare two kinds of positional encoding (PE), *i.e.,* traditional positional encoding (written as 'base') (Zuo et al., 2020) and our FCPE. Moreover, we compare our model with (w/) and without (w/o) multi-scale parts. Meanwhile, we compare two kinds of distribution, *i.e.,* exponential distribution and Weibull distribution.

**Base vs. FCPE.** When transitioning from the base positional encoding to FCPE, we observe consistent performance improvements (2% to 4%) across all datasets. This highlights the effectiveness of FCPE in capturing the cyclic patterns and multi-scale temporal dependencies present in event sequences.

**Multi-scale attention.** Introducing multi-scale attention further enhances predictive accuracy. The model with multi-scale attention consistently outperforms its counterpart without it. This demonstrates the significance of modeling interactions at different temporal scales, which is crucial for capturing complex event dependencies.

**Exponential vs. Weibull.** Comparing the two event time distributions, we find that the Weibull distribution yields better results. This aligns with our theoretical justification that the Weibull distribution can better reflect the varying influences of history on current events. The Weibull distribution, when incorporated into the point process, allows for capturing intricate temporal patterns influenced by past events, whereas the Exponential distribution remains memoryless.

## 4.4 COMPUTATIONAL TIME COSTS

We maintained a consistent configuration by fixing the batch size, embedding dimension, and other hyperparameters such as the learning rate. We compared the time costs of different methods on the two datasets Medications and Providers. The results are shown in Figure 4(a). Our model has lower time costs compared with other baseline transformers. Furthermore, we use extended sequence in the medication dataset and evaluated XTSFormer against its single-scale variant. As illustrated in Figure 4(b), the time discrepancy between XTSFormer and its single-scale counterpart widens with increasing sequence length, underscoring the efficiency of the multi-scale approach.

Table 3: Accuracy results (average $\pm$ std) of ablation study expressed in percentages, where PE means positional encoding.

| PE | Multi-Scale | Distribution | Medications | Providers | Financial | StackOverflow |
|---|---|---|---|---|---|---|
| base | w/o | Exponential | $25.2_{\pm0.3}$ | $36.7_{\pm1.3}$ | $71.1_{\pm1.5}$ | $45.1_{\pm1.5}$ |
| FCPE | w/o | Exponential | $27.8_{\pm0.2}$ | $38.9_{\pm0.9}$ | $72.0_{\pm1.8}$ | $47.0_{\pm0.9}$ |
| base | w/ | Exponential | $28.3_{\pm0.5}$ | $37.9_{\pm1.0}$ | $73.6_{\pm1.6}$ | $46.9_{\pm1.0}$ |
| FCPE | w/ | Exponential | $30.9_{\pm0.6}$ | $38.1_{\pm0.8}$ | $74.8_{\pm2.6}$ | $48.5_{\pm1.0}$ |
| base | w/o | Weibull | $26.8_{\pm0.6}$ | $37.1_{\pm0.8}$ | $73.9_{\pm2.0}$ | $46.5_{\pm1.1}$ |
| FCPE | w/o | Weibull | $28.9_{\pm0.3}$ | $39.6_{\pm0.6}$ | $74.1_{\pm1.9}$ | $47.1_{\pm0.5}$ |
| base | w/ | Weibull | $29.3_{\pm0.6}$ | $40.2_{\pm1.3}$ | $73.5_{\pm1.8}$ | $45.1_{\pm0.6}$ |
| FCPE | w/ | Weibull | $33.5_{\pm0.8}$ | $43.9_{\pm1.3}$ | $77.5_{\pm1.9}$ | $49.4_{\pm1.1}$ |

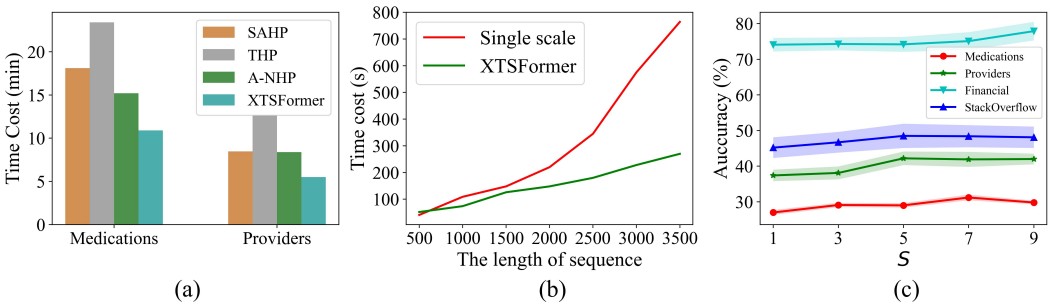

(a)  (b)  (c)

Figure 4: (a) is the time cost of different baselines on two datasets. (b) is the time cost comparison of XTSFormer and its single scale version on Medication. (c) is the comparison of accuracy on various $S$ scales.

## 4.5 SENSITIVITY ANALYSIS

We investigate the parameter sensitivity by varying the largest scale $S \in \{1, 3, 5, 7, 9\}$ and report the accuracy results on four datasets in Figure 4(c). Notably, our method displays sensitivity to the largest scale $S$, as it determines the multi-scale intensity. For instance, when $S = 1$, only a single scale is present, leading to suboptimal performance.

## 5 CONCLUSION AND FUTURE WORKS

This paper introduces a new multi-scale neural point process model called XTSFormer. Our XTS-Former consists of feature-based cycle-aware positional encoding (FCPE) and cross-scale temporal attention within a multi-scale time hierarchy. Specifically, we define the time scale on irregular time event sequences by the merging order of a bottom-up clustering algorithm. The idea is motivated by the fact that events with shorter intervals (at smaller scales) will be merged earlier. We designed a cross-scale attention operation by specifying the key set as nodes in the same scale level. Experiments on two real patient safety datasets and two public datasets demonstrate the superior performance of our proposed XTSformer. In future work, we intend to further analyze and improve our proposed model to enhance its robustness and reliability, particularly in situations with outliers and noisy event data.

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

# A  APPENDIX

## A.1  THEORY OF FCPE

This section delves into the theoretical base of the Feature-based Cycle-aware Positional Encoding (FCPE).

### A.1.1  COMPLIANCE WITH BOCHNER'S THEOREM

Consider the temporal kernel defined as:

$$\mathcal{K}\left(t_a, t_b\right) = \mathcal{P}\left(t_a\right) \cdot \mathcal{P}\left(t_b\right) = \mathcal{F}\left(t_a - t_b\right). \tag{8}$$

This kernel, dictated by its Gram matrix, exhibits translation invariance. Specifically, for any constant $c$:

$$\mathcal{K}\left(t_a, t_b\right) = \mathcal{F}\left(t_a - t_b\right) = \mathcal{K}\left(t_a + c, t_b + c\right). \tag{9}$$

Given that the mapping

$$\mathcal{P} : T \to \mathbb{R}^{d \times 1}$$

is continuous, the kernel meets the condition set by Bochner's Theorem.

**Theorem 2** *(Bochner's Theorem). A kernel that is both continuous and translation-invariant, given by*

$$\mathcal{K}\left(t_a, t_b\right) = \mathcal{F}\left(t_a - t_b\right),$$

*is positive definite if and only if there exists a non-negative measure on $\mathbb{R}$ such that $\mathcal{F}$ is the Fourier transform of this measure.*

### A.1.2  TRANSLATION INVARIANCE OF FCPE

The translation invariance property ensures its consistent performance irrespective of shifts in the input feature, and enhancing its generalizability towards timespan, as proofed below.

$$\begin{aligned}
\mathcal{P}\left(t_a\right) \mathcal{P}\left(t_b\right) &= \sum_{i=1}^{d} \left[\mu_a^i \mu_b^i \cos\left(w_i t_a\right) \cos\left(w_i t_b\right) + \mu_a^i \mu_b^i \sin\left(w_i t_a\right) \sin\left(w_i t_b\right)\right] \\
&= \sum_{i=1}^{d} \mu_a^i \mu_b^i \left[\cos\left(w_i t_a\right) \cos\left(w_i t_b\right) + \sin\left(w_i t_a\right) \sin\left(w_i t_b\right)\right] \\
&= \sum_{i=1}^{d} \mu_a^i \mu_b^i \cos\left(w_i \left(t_a - t_b\right)\right).
\end{aligned} \tag{10}$$

Thus,

$$\mathcal{P}\left(t_a + c\right) \mathcal{P}\left(t_b + c\right) = \sum_{i=1}^{d} \mu_a^i \mu_b^i \cos\left(w_i \left(t_a - t_b\right)\right) = \mathcal{P}\left(t_a\right) \mathcal{P}\left(t_b\right), \tag{11}$$

which implies that $\mathcal{P}\left(t_a\right) \mathcal{P}\left(t_b\right)$ only influenced by the timespan $t_a - t_b$.

## A.2  EXPERIMENTS

### A.2.1  EXPERIMENTAL SETUP

All experiments were conducted using PyTorch on a server equipped with NVIDIA A100 80GB Tensor Core GPU. During training, we set the initial learning rate in the range of [0.0001, 0.001] and the weight decay within [0, 0.0001] for all datasets, respectively, with the Adam optimizer (Kingma & Ba, 2015). The dimension of embedding, the epoch number, the learning rate, and the weight decay of our methods, respectively, are set as 256, 200, $1 \times 10^{-3}$ and $1 \times 10^{-4}$. In addition to directly adding the FCPE temporal embedding to the non-temporal feature embedding, we also experimented with concatenating them, setting the dimension for each at 128. We employ an early stopping patience of 25, terminating training if there's no decrease in loss over 25 epochs. To avoid the over-fitting issue on the datasets with limited subjects, in all experiments, we repeat the 5 times with different random seeds on all datasets for all methods. We finally report the average results and the corresponding standard deviation (std).

Table 4: The statistic of all datasets.

| Dataset | # of Event Types | # of Events | Length of Sequence | | | # of Sequence | | |
|---|---|---|---|---|---|---|---|---|
| | | | Min | Mean | Max | Train | Validation | Test |
| Medications | 86 | 355,490 | 10 | 70 | 878 | 4,064 | 508 | 508 |
| Providers | 48 | 704,496 | 10 | 13 | 48 | 45,010 | 5,626 | 5,626 |
| Financial | 2 | 414,800 | 829 | 2,074 | 3,319 | 160 | 20 | 20 |
| StackOverflow | 22 | 480,413 | 41 | 72 | 736 | 4,777 | 530 | 1,326 |

### A.2.2 DATASETS DESCRIPTION

**Medications** dataset originates from our university's hospital and encompasses approximately 5,000 patient encounters. Each encounter, treated as a sequence, chronologically details the patient's medication records, spanning 86 distinct subcategories. The medication timelines cover pre-operations, intra-operations, and post-operations periods.

**Providers** dataset, sourced from our university's hospital, parallels the structure of the Medications dataset, containing around 55,000 patient encounters. Each encounter is treated as a sequence detailing the patient's provider interactions in chronological order, with a breakdown into 48 distinct provider classifications by function. These interactions span the pre-operations, intra-operations, and post-operations periods.

**Financial** (Du et al., 2016) dataset comprises high-frequency transaction records for a stock in a single day. Each entry is timestamped (in milliseconds) and indicates an action (buy/sell). While the data forms a continuous sequence, it is partitioned for training and testing purposes. The goal is to predict the timing of a particular action.

**StackOverflow** (Leskovec & Krevl, 2014) is a Questions and Answers platform that incentivizes user engagement through badge rewards, with users potentially receiving the same badge multiple times. We analyze a two-year data span, representing each user's badge acquisition as a sequence, where each event denotes a specific badge receipt.

