# OpenReview forum: "XTSFormer: Cross-Temporal-Scale Transformer for Irregular Time Event Prediction"
_ICLR.cc/2024/Conference — Submitted to ICLR 2024_

### Official Review · Reviewer_AZAP · 2023-10-30

**Soundness:** 3 good
**Presentation:** 3 good
**Contribution:** 3 good
**Rating:** 8
**Confidence:** 4

**Summary:**

This paper proposes a new model called XTSFormer for irregularly timed event data, and highlights its superior performance compared to baseline methods. The XTSFormer includes a unique Feature-based Cycle-aware Positional Encoding and a hierarchical multi-scale temporal attention mechanism to address these challenges.

**Strengths:**

1. The proposed PCPE incorporates event type information. It is reasonable to use type information for better embedding.
2. The proposed multi-scale attention is novel and useful.
3. The use of Weibull distribution improves performance.

**Weaknesses:**

1. The proposed FCPE seems to be Mercer time embedding [1] parameterized by event type. It would be great to present the comparison with Mercer time embedding.
2. The ablation studies on the NLL/RMSE with FCPE, attention, and weibull distribution should be presented.
3. The use of event type prediction somehow hinders the models from being a generative model. Can authors run experiments on using the event type prediction similar to ANHP[2]?

[1] Self-attention with Functional Time Representation Learning, NeurIPS 2019.

[2] Transformer Embeddings of Irregularly Spaced Events and Their Participants. ICLR 2022.

**Questions:**

See weakness

---

> ### Author Response · Authors · 2023-11-17
> **Reply to Reviewer AZAP**
>
> We appreciate the great comments and suggestions! We used **W** to represent the weaknesses listed.
>
> ## 1. Comparison with Mercer Time Embedding (W1)
>
> Thank you for bringing up this related work on Mercer Time Embedding [1]. This is a great insight. As suggested, we have added experimental comparisons.
>
> We replaced the FCPE in XTSFormer with Mercer Time Embedding and conducted experiments on four datasets. The results in the following table show that our method has slightly better performance, which may be due to the event type feature-based encoding.
>
> |        | Medications |      | Providers |      | Financial |      | StackOverflow |      |
> |:------:|:-----------:|:----:|:---------:|:----:|:---------:|:----:|:-------------:|:----:|
> |        |   Accuracy (%)  | RMSE |  Accuracy (%) | RMSE |  Accuracy (%) | RMSE |    Accuracy (%)   | RMSE |
> | Mercer |     31.5    | 1.45 |    41.2   | 2.49 |    76.9   | 0.87 |      46.3     | 4.38 |
> |  FCPE  |     33.5    | 1.12 |    43.9   | 2.33 |    77.5   | 0.88 |      49.4     | 4.21 |
>
>
> [1] Self-attention with Functional Time Representation Learning, NeurIPS 2019.
>
>
> ## 2. Ablation Studies on NLL/RMSE (W2)
>
> This is a great point. We have conducted comprehensive ablation studies to fill in the missing details as below, and we will include these studies in our revised manuscript.
>
> ### The NLL of ablation study results:
>
> |  PE  | Multi-scale | Distribution | Meications | Providers | Financial | StackOverflow |
> |:----:|:-----------:|:------------:|:----------:|:---------:|:---------:|:-------------:|
> | base |     w/o     |  Exponential |    3.48    |    2.21   |    2.10   |      2.88     |
> | FCPE |     w/o     |  Exponential |    2.99    |    2.15   |    2.07   |      2.41     |
> | base |      w/     |  Exponential |    2.47    |    1.92   |    1.60   |      2.42     |
> | FCPE |      w/     |  Exponential |    2.20    |    1.81   |    1.17   |      2.10     |
> | base |     w/o     |   Weilbull   |    2.59    |    2.30   |    2.35   |      2.31     |
> | FCPE |     w/o     |   Weilbull   |    2.34    |    2.10   |    2.11   |      2.09     |
> | base |      w/     |   Weilbull   |    2.31    |    1.89   |    1.59   |      2.02     |
> | FCPE |      w/     |   Weilbull   |    2.23    |    1.75   |    1.18   |      1.89     |
>
> ### The RMSE of ablation study results:
>
> |  PE  | Multi-scale | Distribution | Meications | Providers | Financial | StackOverflow |
> |:----:|:-----------:|:------------:|:----------:|:---------:|:---------:|:-------------:|
> | base |     w/o     |  Exponential |    1.89    |    3.62   |    1.44   |      5.85     |
> | FCPE |     w/o     |  Exponential |    1.80    |    3.05   |    1.45   |      5.50     |
> | base |      w/     |  Exponential |    1.75    |    2.48   |    1.01   |      4.39     |
> | FCPE |      w/     |  Exponential |    1.15    |    2.40   |    0.92   |      4.22     |
> | base |     w/o     |   Weilbull   |    1.90    |    3.50   |    2.42   |      5.51     |
> | FCPE |     w/o     |   Weilbull   |    1.82    |    2.88   |    0.89   |      4.39     |
> | base |      w/     |   Weilbull   |    1.65    |    2.51   |    1.15   |      4.30     |
> | FCPE |      w/     |   Weilbull   |    1.12    |    2.33   |    0.88   |      4.21     |
>
> The results from our ablation study demonstrate the robust performance of our Feature-based Cycle-aware Positional Encoding (FCPE), multi-scale attention, and Weibull distribution across various datasets. Notably, the integration of FCPE and multi-scale attention consistently yields lower RMSE values, underlining the efficacy of these components in our model. For example, in the 'Medications' dataset, the combination of FCPE with multi-scale attention and Weibull distribution significantly reduces RMSE to 1.12 from a base RMSE of 1.90. This trend is observed across all datasets, confirming the advantages of our approach as discussed in section 4.3 of our paper.
>
> ## 3. Using ANHP Setup (W3)
> This is a great point! We acknowledge that our current configuration is for next-event prediction. Due to the need for precomputing hierarchical clustering tree, it is not the best fit for event sequence generation, as in the ANHP [2] model. We will work on customizing our model for event sequence generation (autoregressive prediction).
>
> [2] Transformer Embeddings of Irregularly Spaced Events and Their Participants. ICLR 2022.

---

> > ### Comment · Reviewer_AZAP · 2023-11-22
> >
> > Thank you for the reply.
> >
> > The comparison between FCPE and Mercer and the results on NLL/RMSE are interesting. The authors have addressed most of my concerns. I raised my score.
> >
> > Overall, the proposed FCPE and hierarchical architecture is novel and useful. However, the non-autoregressive design still will be my biggest concern for the practicability of the proposed method in Irregular Time Series as the event prediction is only one part of Irregular Time Series processing.

---

> > > ### Author Response · Authors · 2023-11-22
> > >
> > > Thank you for your kind feedback. We fully agree that the autoregressive prediction design is very important. We are still working on it now and it will take a little bit time. Hopefully this will be completed soon.

---

### Official Review · Reviewer_TWtn · 2023-11-02

**Soundness:** 3 good
**Presentation:** 2 fair
**Contribution:** 2 fair
**Rating:** 3
**Confidence:** 3

**Summary:**

The paper proposed a method for predicting the time and type of future events based on a historical event sequence. The model has two main components:
• A feature-based cycle-aware positional encoding (FCPE) that encodes both the cyclic and semantic information of event times into a vector representation.
• A cross-temporal-scale transformer that captures the multi-scale temporal dependencies among events using a hierarchical clustering algorithm and a cross-scale attention mechanism.

**Strengths:**

The paper addresses an interesting issue of event prediction (time and type) using irregular and multi-scale temporal data.
The paper addresses a significant problem, namely event prediction and event timing in the presence of irregularities. This has not been addressed before, which is more challenging than a typical forecasting task.
The paper includes a comprehensive ablation study of each component, and the experiment was conducted on a set of real-world data.

**Weaknesses:**

The model component lacks novelty, which can be seen as a limitation. Nonetheless, the paper's exploration of event prediction within irregular sequences is an interesting topic.

The paper never addresses state-of-the-art irregular models recently introduced, including ODE-based models [1-4], which is critical to be added as one of the baseline models.
Recent models [2,3,4] can handle irregular time series more effectively without the need for data grouping based on temporal scale. A comparison with ODE-based models is essential.

The paper is generally well-written. However, the model component is difficult to understand; many details are in the appendix.
There are issues in the organization of Section 3, particularly in understanding the connection between different modules. For instance, Section 3.2.4, which outlines the overall model architecture, must be presented at the beginning of Section 3, followed by a breakdown of individual module components. The excessive use of subsubsections with limited content also hampers the paper's clarity. Structural improvements are needed to enhance the overall readability.


[1] Chen, R.T., Rubanova, Y., Bettencourt, J. and Duvenaud, D.K., 2018. Neural ordinary differential equations. Advances in neural information processing systems, 31.
[2] Kidger, P., Morrill, J., Foster, J. and Lyons, T., 2020. Neural controlled differential equations for irregular time series. Advances in Neural Information Processing Systems, 33, pp.6696-6707.
[3] Rubanova, Y., Chen, R.T. and Duvenaud, D.K., 2019. Latent ordinary differential equations for irregularly-sampled time series. Advances in neural information processing systems, 32.
[4] Weerakody, P.B., Wong, K.W., Wang, G. and Ela, W., 2021. A review of irregular time series data handling with gated recurrent neural networks. Neurocomputing, 441, pp.161-178.

**Questions:**

There are notable issues with the clarity and novelty of the proposed model. Why has ODE based models not considered for comparison?

Why is the use of hierarchical classification needed? It raises concerns due to the data preprocessing required.

---

> ### Author Response · Authors · 2023-11-17
> **Reply to Reviewer TWtn**
>
> Thank you for the insightful comments. Please see our responses below. We used **Q** and **W** to represent the questions and weaknesses listed, respectively.
>
> ## 1. Comparison with ODE-based Models (Q1)
>
> We recognize the importance of comparing our model against Neural ODE-based models.
>
> It is true that Neural ODE-based models are popular methods in handling irregular time series [1,2,3,4]. However, its common version [1-4] captures random variables as a continuous-time function (e.g., temperature over a time period), which **cannot be directly applied to discrete event sequences** (as the events do not occur at every time point in the continuous time domain, unlike temperature variable). After a literature review, we found two papers that customized the neural ODE models for discrete event sequences: Neural Jump Stochastic Differential Equations (NJSDE) [6], which enhances the Neural ODE framework with a stochastic process on jumps between discrete events; and [7] that applies Neural ODE to compute the likelihood of Spatio-Temporal Point Processes (ODETPP). For a fair comparison, we modified ODETPP by removing its spatial component (only focusing on time).
>
> We conducted an experimental comparison between our method and existing Neural ODE-based TPPs [6] and [7] on two public datasets. Our experimental results below demonstrate that XTSFormer has better prediction accuracy over neural ODE-based TPP on two public datasets.
>
> |           | Financial |          |      |      | StackOverflow |          |      |      |
> |:---------:|:---------:|:--------:|:----:|:----:|:-------------:|:--------:|:----:|:----:|
> |           |  Accuracy (%) | F1-score (%)| RMSE |  NLL |    Accuracy (%)   | F1-score (%)| RMSE |  NLL |
> |   NJSDE [6]  |    72.0   |   74.1   | 0.93 | 1.18 |      47.3     |   34.2   | 4.33 | 1.96 |
> |   ODETPP [7]  |    69.1   |   65.9   | 2.20 | 2.07 |      44.4     |   33.1   | 5.46 | 2.52 |
> | XTSFormer |    77.5   |   78.8   | 0.88 | 1.18 |      49.4     |   35.0   | 4.21 | 1.89 |
>
> [1] Chen, R.T., Rubanova, Y., Bettencourt, J. and Duvenaud, D.K. Neural ordinary differential equations. Advances in neural information processing systems, 2018.
> [2] Kidger, P., Morrill, J., Foster, J. and Lyons, T.. Neural controlled differential equations for irregular time series. Advances in Neural Information Processing Systems, 2020.
> [3] Rubanova, Y., Chen, R.T. and Duvenaud, D.K.. Latent ordinary differential equations for irregularly-sampled time series. Advances in neural information processing systems, 2019.
> [4] Weerakody, P.B., Wong, K.W., Wang, G. and Ela, W., 2021. A review of irregular time series data handling with gated recurrent neural networks. Neurocomputing, 441, pp.161-178.
> [5] Brillinger D R. Time series, point processes, and hybrids. Canadian Journal of Statistics, 1994.
> [6] Jia J, Benson A R. Neural jump stochastic differential equations. Advances in Neural Information Processing Systems, 2019.
> [7] Chen R T Q, Amos B, Nickel M. Neural Spatio-Temporal Point Processes. International Conference on Learning Representations, 2021.
>
>
> ## 2. Why Hierarchical Clustering (Q2)
> This is a great question. We use hierarchical clustering on irregular-time events in the time domain to capture the multi-scale temporal patterns on such event sequences. Such multi-scale temporal event patterns are motivated by real-world applications. For example, in the electronic health record (EHR) data we received from collaborators in a hospital, irregular time medication events for the same patient are multi-scale in nature. Medications intra-operation (in operating rooms) are often every few minutes (**fine granularity**) while medication events pre- or post-operations are often every few hours (**coarse granularity**). Our domain collaborator in clinical practice also highlights the importance of such multi-scale patterns. Thus, we propose a method to capture the multi-scale temporal patterns inherent in such real-world irregular-time event datasets. We will add such discussions as you suggested. Thank you for bringing this up!
>
> We acknowledge that the idea poses preprocessing costs, and we are planning to explore an end-to-end learnable approach for hierarchical clustering for multi-scale temporal patterns. This will be explored in future work.
>
> ## 3. Improvements in Clarity and Organization (W3)
>
> We fully agree with the points regarding the presentation and organization issues. We believe these issues can be fixed. We are working on that now.

---

> ### Author Response · Authors · 2023-11-22
> **Reminder of Response to Author's Rebuttal**
>
> Dear Reviewer TWtn,
>
> Thanks for your suggestions. We have carefully addressed your comments on comparison with existing ODE-based models. We would greatly appreciate it if you could review our responses and consider if they might prompt an adjustment in your scoring.
>
> Additionally, in response to your feedback on presentation and organization, we have restructured Section 3 to provide a clearer and more coherent introduction to the overall model architecture. We are committed to continuously revising the paper further to enhance its clarity and readability, following your insightful suggestions.
>
> Your feedback is very valuable to us, and we eagerly await any further insights you might offer.

---

> > ### Comment · Reviewer_TWtn · 2023-11-23
> >
> > Thanks for the additional experiments and detailed comments. Much appreciated. However, the experiment is incomplete as there are several newer ODE models. I stand by my score.

---

### Official Review · Reviewer_cLDM · 2023-11-07

**Soundness:** 2 fair
**Presentation:** 2 fair
**Contribution:** 1 poor
**Rating:** 3
**Confidence:** 4

**Summary:**

The cross-temporal-scale transformer (XTSFormer) is proposed as a solution for unevenly timed event data. The approach is built around two key components: a unique Feature-based Cycle-aware Positional Encoding (FCPE) that captures the cyclical character of time and a hierarchical multi-scale temporal attention mechanism. A bottom-up clustering technique determines these scales. Extensive testing on a variety of real-world datasets demonstrates that our XTSFormer exceeds various baseline methods in prediction performance.

**Strengths:**

It is the first work to take into account multi-scale features in neural TPPs.

They introduce a novel time positional encoding, the Feature-based Cycle-aware Positional Encoding (FCPE), which incorporates both feature and cyclical information. This approach strengthens our model’s capacity to capture complex temporal patterns.

They design a cross-scale attention operation within the multi-scale hierarchy and compare the time cost with default all-pair attention.

**Weaknesses:**

(1) It is not so clear why multi-scale patterns are so important for event sequences. In introduction, there is no explanation about the benefits of multi-scale features.

(2) Multi-scale temporal attention mechanism has been widely published in previous papers [1][2]. Cycle-aware Time Embedding has been proposed in [3]. Please describe the difference between your paper and these published papers. The novelty is limited.

(3) Eq.6 and Eq. 7 are unnecessary. They are the commonsense of attention mechanism and MLP.

(4) Section 3.2.3 is so simple. Please add more analysis about Flops and memory cost of your model.

(5) Each equation should be followed of commas or full stop. Please clarify it.

(6) Some formulas are not clearly defined, such as the missing definition of P(ti) in Eq. (5)

(7) The writing of the paper can be improved as there are many typos.


[1]Hu H, Dong S, Zhao Y, et al. Transrac: Encoding multi-scale temporal correlation with transformers for repetitive action counting[C]//Proceedings of the IEEE/CVF Conference on Computer Vision and Pattern Recognition. 2022: 19013-19022.

[2]Dai R, Das S, Kahatapitiya K, et al. MS-TCT: multi-scale temporal convtransformer for action detection[C]//Proceedings of the IEEE/CVF Conference on Computer Vision and Pattern Recognition. 2022: 20041-20051.

[3]Dikeoulias I, Amin S, Neumann G. Temporal knowledge graph reasoning with low-rank and model-agnostic representations[J]. arXiv preprint arXiv:2204.04783, 2022.

**Questions:**

Please see weakness above.

**Details Of Ethics Concerns:**

No.

---

> ### Author Response · Authors · 2023-11-17
> **Reply to Reviewer cLDM (Part 1/2)**
>
> Thank you for your constructive feedback and for recognizing the novel aspects of our XTSFormer model. We value your insights and are eager to address the concerns raised to enhance the clarity and impact of our work. We used **W** to represent the weaknesses listed.
>
> ## 1. Importance of Multi-scale Patterns in Event Sequences (W1)
> Thanks for the comment. Our idea of multi-scale temporal event patterns is motivated by rea-world applications. For example, in the electronic health record (EHR) data we received from collaborators in a hospital, irregular time medication events for the same patient are multi-scale in nature. Medications administered intra-operation (in operating rooms) are often every few minutes (fine granularity), while medication events pre- or post-operations are often every few hours (coarse granularity). Our domain collaborator in clinical practice also highlights the importance of such multi-scale patterns. Thus, we propose a method to capture the multi-scale temporal patterns inherent in such real-world irregular-time event datasets. We will add such explanations in the introduction as you suggested.
>
> ## 2. Comparison with Multi-scale Temporal Attention and Cycle-aware Time Encoding (W2)
> Thank you for the great suggestions to clarify novelty against existing works on multi-scale temporal attention and cycle-aware time embedding. We have now discussed these two aspects one by one.
>
> First, the two works on multi-scale temporal attention (e.g., **TransRAC [1]**, **MS-TCT [2]**) are designed for action detection in video data which are **regular time series**. For instance, MS-TCT uses temporal convolutional layers to process time series data, which cannot be directly applied to irregular time event data in our case. In contrast, our XTSFormer focuses on multi-scale temporal attention on **irregularly timed event data** in a neural temporal point process framework. We will cite these two papers in related work and add discussions on the differences in our draft.
>
> Second, we carefully reviewed the existing cycle-aware time embedding paper [3] you suggested.
> The Multi-Recurrent Cycle-Aware (MRCA) encoding [3] decomposes time into multiple cycles like seasons, months, weeks, and days to capture recurring temporal patterns. However, we found several differences as summarized below:
>
> MRCA directly encodes the time into several **fixed cycles**, e.g., daily, weekly, monthly, which may not be sufficient for datasets where cycles are not well-defined. In contrast, our Feature-based Cycle-aware Positional Encoding (FCPE) has **learnable cycles representation**. It is specifically tailored for irregular temporal data. Although cycle-aware time positional encodings are not new (see [4,5] below), we added the idea of semantic-feature-based learnable cycles. Specifically, the learnable cycles in our FCPE depend on **event-specific semantic features**. This is motivated by real-world applications where different types of events may show different cyclic patterns. For example, some types of medications are taken every few minutes, while others may be every few hours or daily. Another difference is that the existing work of MRCA [3] lacks **time translation invariance**. This means it is unable to assure steady performance across the time domain, particularly when facing shifts in temporal features.
>
> We also added experimental comparisons of our feature-based cycle-aware time positional encoding (FCPE) against existing basic cycle-aware time position encoding methods, including [4] (we call base below), and Mercer encoding [5] brought up by another reviewer (i.e., positional encoding without event feature parameterization). The results are as follows:
>
>
> |        | Medications |      | Providers |      | Financial |      | StackOverflow |      |
> |:------:|:-----------:|:----:|:---------:|:----:|:---------:|:----:|:-------------:|:----:|
> |        |   Accuracy (%) | RMSE |  Accuracy (%) | RMSE |  Accuracy (%) | RMSE |    Accuracy (%)   | RMSE |
> |  base  |     29.3    | 1.65 |    40.2   | 2.51 |    73.5   | 1.15 |      45.1     | 4.30 |
> | Mercer |     31.5    | 1.45 |    41.2   | 2.49 |    76.9   | 0.87 |      46.3     | 4.38 |
> |  FCPE  |     33.5    | 1.12 |    43.9   | 2.33 |    77.5   | 0.88 |      49.4     | 4.21 |
>
> The results show that our FCPE has slightly better prediction accuracy.
>
>
> [1]Hu H, Dong S, Zhao Y, et al. Transrac: Encoding multi-scale temporal correlation with transformers for repetitive action counting. CVPR 2022.
>
> [2]Dai R, Das S, Kahatapitiya K, et al. MS-TCT: multi-scale temporal convtransformer for action detection. CVPR 2022.
>
> [3]Dikeoulias I, Amin S, Neumann G. Temporal knowledge graph reasoning with low-rank and model-agnostic representations.  Workshop on representation learning for NLP 2022.
>
> [4] Zuo, Simiao, et al. "Transformer hawkes process." ICML 2020.
>
> [5] Self-attention with Functional Time Representation Learning, NeurIPS 2019.

---

> ### Author Response · Authors · 2023-11-17
> **Reply to Reviewer cLDM (Part 2/2)**
>
> ## 3. Time and Memory Cost Analysis (W4)
>
> Thank you for the suggestions. We have added a detailed analysis of our approach's time and memory costs as below.
>
> ### Time cost analysis (FLOPs):
> Assume the number of input temporal points is $L$, the batch size is $B$, the number of heads is $h$, and the hidden dimensions are $d$. In our attention computation, the query matrix dimensions is represented by $Q \in \mathbb{R}^{B \times h \times L \times d}$. For each query point $\mathcal{Q}_i$  (where $1 \leq i \leq L$), its selective key set size is $M$ (dependent on the threshold in each level). Thus, the key matrix results from concatenating key sets of all query points, yielding a dimension of $\mathbb{R}^{B \times h \times L \times M \times d}$. Consequently, the FLOPs of the attention computation with our approach is $B \cdot h \cdot L \cdot M \cdot d$, with the key set size $M$ scaling $O(log L)$. This results in a significantly more efficient attention computation Flops, specifically $B \cdot h \cdot L \cdot \log L \cdot d$, compared to the vanilla transformer [6] computation of $B \cdot h \cdot L^2 \cdot d$.  For example, when $B=64, h= 4, L=10000, d=256$, the Flops per attention layer = $2\times 64 \times 4 \times 10000 \times \log 10000 \times 256 \approx 17.4$ billion FLOPs.
>
> ### Memory cost analysis:
> We analyze the memory costs of the model theoretically. We use the same math variable definition as above. We use $\mathcal{K}_i$ to denote the key set for the $i$-th query point and key set size is $M$. The memory costs of our approach are dominated by the hierarchical attention layer, since we need to store the key set $\mathcal{K}_i$ for $L$ query points, and each key set has the dimension $M \cdot d$. Thus, the total memory cost is $O(B\cdot h  \cdot d \cdot L \cdot M)$ per layer, where $M=\log L$.  Given $B=64, h= 4, L=10000, d=256$, with floating32 precision, the memory cost is  $64\times 4\times 256 \times 10000 \times log 10000 \times 4 \approx 32.4$ GB.
>
> Additionally, to evaluate the efficiency of our method on long event sequences, we have conducted further computational experiments using a synthetic dataset. This dataset comprises $64$ event data sequences, each with a length of $100,000$ events. Each event in a sequence is a pair: an event type and its corresponding time point. The event type is a discrete variable ranging from ${0, 1, 2, \dots, 9}$. Each event type exhibits a unique temporal occurrence pattern, modeled using a Poisson distribution. Given an event type $i$, the time point of its occurrence is sampled from an exponential distribution with rate parameter $\lambda_i$. This is given by:$P(T = t | \lambda_i) = \lambda_i e^{-\lambda_i t}$, where $T$ is the time between events. To introduce variability and simulate the irregularities observed in real-world event sequences, we perturb these time intervals by adding Gaussian noise $\epsilon \sim \mathcal{N}(0, \sigma^2)$ where $\sigma$ is a small standard deviation. Based on preliminary testing and to ensure that the perturbations were subtle yet meaningful, we set $\sigma = 0.1\times \text{mean of the sampled time interval}$.
>
> In our experiments, we set the batch size $B=1$ and the hidden dimension $d=32$. We progressively increased the sequence length and recorded the time cost (in seconds) per epoch. The results, which we plan to include in the final version of the paper, are as follows (OOM indicates 'out of memory'):
>
> |  Length of Sequence | 10,000 | 20,000 | 30,000 | 40,000 | 50,000 | 60,000 | 70,000 | 80,000 |
> |:-------------------:|:------:|:------:|:------:|:------:|:------:|:------:|:------:|:------:|
> | Vanilla Transformer [6] |   2.9  |   7.9  |  12.5  |  24.3  |  39.1  |  58.8  |  83.7  |   OOM  |
> |      XTSFormer      |   3.1  |   4.5  |   6.6  |  12.1  |  13.9  |  16.7  |  20.8  |  25.3  |
>
> The experimental results clearly demonstrate that our XTSFormer model is more efficient than the Vanilla Transformer.
>
> [6] Vaswani, Ashish, et al. "Attention is all you need." Advances in neural information processing systems 30 (2017).
>
> ## 4. Improvements in Writing and Technical Presentation (W3, W5, W6, W7)
>
> Thank you for the great catch on writing issues concerning equations and typos. We have fixed those in our revised draft.

---

> > ### Comment · Reviewer_cLDM · 2023-11-21
> > **Response to Authors**
> >
> > Thank you for your response. I still have concerns about the novelty of this paper. The main framework (called Cross-Temporal-Scale Transformer) can be easily found in other published papers. As the author says, the regular time series and irregularly timed event data have great difference, but I disagree with the argument that the distribution of data has impact on the framework. Your framework will be better if you can tackle with both regular and irregular conditions.

---

> ### Author Response · Authors · 2023-11-21
> **Response to reviewer view that distribution of data has no impact on the framework**
>
> We acknowledge that a model that can tackle both regular and irregular conditions will be ideal. Technically, we can of course run the framework on both regular and irregular sequences as long as the input format looks similar.
> We disagree with the reviewer, however, on the view that the distribution of data has no impact on the framework. We urge the reviewer to be careful to draw such a conclusion since there is a rich literature highlighting the difference in framework for time series forecasting and temporal point process. Statistically, they are different stochastic processes. In fact, not all regular-time series transformers are applicable to our irregular-time event sequence problems (including the two methods recommended by the reviewer). As said in [6], **Temporal point processes (TPPs)** are principled mathematical tools for the modeling and learning of asynchronous event sequences, which capture the instantaneous happening rate of the events and the temporal dependency between historical and current events. So **TPPs are the dominant paradigm for modeling sequences of events happening at irregular intervals [18]**. Our model proposed a neural TPP method. In fact, there is rich literature from major machine learning and AI conferences (e.g., AAAI, ICLR, ICML) highlighting that **there is a big gap between regular data (like time series) and irregular data (like even data, TPPs)**. For example, as stated in [1], "Unlike other sequential data such as time series, event sequences tend to be asynchronous, which means time intervals between events are just as important as the order of them to describe their dynamics". Furthermore, as claimed in [2], compared with regularly-spaced time series, TPPs are different since they are irregularly-spaced time sequences. As stated in [3], the time series and event data (TPPs) are different domains. As claimed in [4], the true distribution of events is never known and the performance depends on the design of the stochastic process. As claimed in [5], the irregularity and asynchronicity of event sequences pose a challenge. Similar arguments can also be found in [6-20].
>
> In addition, can you reviewer provide evidence that the same “framework (called Cross-Temporal-Scale Transformer) can be easily found in other published papers”? Please kindly show us an example of paper that has exactly the same method and we will make sure we compare with them.
>
> Could the reviewer comment on our response to your concern about time complexity and cost analysis?
>
> References:
> [1] Zuo, Simiao, et al. "Transformer Hawkes Process". ICML 2020.
>
> [2] Bae, Wonho, et al. "Meta Temporal Point Processes". ICLR 2023.
>
> [3] Ding, Fangyu, et al. "c-NTPP: Learning Cluster-Aware Neural Temporal Point Process". AAAI 2023.
>
> [4] Pan, Zhen, et al. "A Variational Point Process Model for Social Event Sequences". AAAI 2020.
>
> [5] Zhang, Qiao, et al. "Self-Attentive Hawkes Process". ICML 2020.
>
> [6] Yan, Junchi, et al. "Modeling and Applications for Temporal Point Processes". KDD 2019.
>
> [6] Li, Shuang, et al. "Learning Temporal Point Processes via Reinforcement Learning". NeurIPS 2018.
>
> [7] Shou, Xiao, et al. "Influence-Aware Attention for Multivariate Temporal Point Processes". CLeaR 2023
>
> [8] Boyd, Alex, et al. "Probabilistic Querying of Continuous-Time Event Sequences". AISTATS 2023.
>
> [9] Gracious, Tony, et al. "Dynamic Representation Learning with Temporal Point Processes for Higher-Order Interaction Forecasting". AAAI 2023.
>
> [10] Wang, Qingmei, et al. "Hierarchical Contrastive Learning for Temporal Point Processes". AAAI 2023.
>
> [11] Yang, Chenghao, et al. "Transformer Embeddings of Irregularly Spaced Events and Their Participants". ICLR 2022.
>
> [12] Yuan, Yuan, et al. "Spatio-temporal Diffusion Point Processes". KDD 2023.
>
> [13] Chen, T. Q. , et al. "Neural Spatio-Temporal Point Processes".  ICLR 2021.
>
> [14] Zhang, Yunhao, et al. "Learning Mixture of Neural Temporal Point Processes for Multi-dimensional Event Sequence Clustering". IJCAI 2022.
>
> [15] Du, Nan, et al. "Recurrent Marked Temporal Point Processes: Embedding Event History to Vector". KDD 2016.
>
> [16] Mei, Hongyuan, et al. "The Neural Hawkes Process: A Neurally Self-Modulating Multivariate Point Process". NeurIPS 2017.
>
> [17] Omi, Takahiro, et al. "Fully Neural Network based Model for General Temporal Point Processes". NeuIPS 2019.
>
> [18] Shchur, Oleksandr, et al. "Intensity-Free Learning of Temporal Point Processes". ICLR 2020.
>
> [19] Shchur, Oleksandr, et al. "Neural Temporal Point Processes: A Review". IJCAI 2021.
>
> [20] Upadhyay, Utkarsh , et al. "Deep Reinforcement Learning of Marked Temporal Point Processes". NeurIPS 2018.

---

> > ### Comment · Reviewer_cLDM · 2023-11-22
> > **Response to Authors**
> >
> > Thanks for your response. The novelty of "cross-scale transfromer" has been published in ICCV 2021 [1], and this paper has been cited for many times. I think you just modified the structure of transformer in temporal dimension and deploy it into Irregular Time Event Prediction. Therefore, I still have concerns about the novelty of this paper.
> >
> > I don't mean that the distribution of data has no impact on the framework. I just give a suggestion that your framework can perform well in more generalized condition.
> >
> >
> > [1] Chen C F R, Fan Q, Panda R. Crossvit: Cross-attention multi-scale vision transformer for image classification[C]//Proceedings of the IEEE/CVF international conference on computer vision. 2021: 357-366.

---

> ### Author Response · Authors · 2023-11-22
>
> Dear Reviewer cLDM,
>
> Thank you for your elaboration. We carefully read the related work [1] (ICCV 2021) you brought up. We agree with you that the general philosophy shares some similarity in that both methods address efficiency through coarse-scale and fine-scale attention.  Saying that, we still see some differences that can be important.
>
> Specifically, Cross-ViT in paper [1] uses two attention branches (one for **small patches**, and the other for **coarse patches**) within a regular image grid. Because of the **regular grid**, different patch-scales are naturally **pre-existing (well-defined)**. This is similar to other multi-scale attention for time series data (e.g., Pyraformer in our references). In contrast, in our irregular time event sequence, the scales are often **implicit and need to be established**, by preprocessing with hierarchical time clustering (our current approach) or more importantly by learning such multi-scale temporal structure (that we plan to explore in future works). Therefore, we did not just modified the structure from [1]. We can add these discussion in a revised version.
>
> We have spent quite some time in addressing your comments, including time complexity analysis and FLOPs and experimental results. Could you consider raising your score?
>
> For your suggestion of a general method for different scenarios, we agree that it is a good to be general. It is just that most event prediction problems are irregular time in nature. But thank you for bringing this up.

---

### Official Review · Reviewer_XqYU · 2023-11-09

**Soundness:** 3 good
**Presentation:** 3 good
**Contribution:** 2 fair
**Rating:** 6
**Confidence:** 3

**Summary:**

The paper presents a natural extension of the Temporal Point Process framework. The idea is to use data split to improve the learning. In particular, the authors propose a specific way to cluster event sequences and calculate modified attention based on such a clustering.

**Strengths:**

- Interesting idea and delivered algorithm
- Reasonable number of datasets in the comparison
- Time cost analysis performed. It shows the superiority of the considered approach for long sequences.

**Weaknesses:**

- The proposed approach can limit the performance for complex long-range dependencies
- Most alternative approaches consider a different structure for event type and intensity prediction, which makes a comparison not 100% fair (while I consider it not a big problem)
- No comparison to other clustering approaches for TPP [1]. However, it is also tolerable, given that the paper came out in 2023. Also lack of comparison with clustering based on embeddings of tokes from NLP

1. Ding, Fangyu, Junchi Yan, and Haiyang Wang. "c-NTPP: learning cluster-aware neural temporal point process." Proceedings of the AAAI Conference on Artificial Intelligence. Vol. 37. No. 6. 2023.

**Questions:**

- Can you compare your approach to c-NTPP?
- Can you compare to other Efficient transformers based on grouping tokens into clusters [2, 3]?

2. Vyas, Apoorv, Angelos Katharopoulos, and François Fleuret. "Fast transformers with clustered attention." Advances in Neural Information Processing Systems 33 (2020): 21665-21674.
3. Wang, Ningning, et al. "Clusterformer: Neural clustering attention for efficient and effective transformer." Proceedings of the 60th Annual Meeting of the Association for Computational Linguistics (Volume 1: Long Papers). 2022.

---

> ### Author Response · Authors · 2023-11-17
> **Reply to Reviewer XqYU**
>
> Thank you for the great suggestions on additional comparisons. We used **Q** and **W** to represent the questions and weaknesses listed, respectively.
>
> ## 1. Comparison with c-NTPP (Q1, W2, W3)
> Thank you for bringing up the recent work on c-NTPP (Cluster-Aware Neural Temporal Point Process), published in AAAI 2023. We compared it with our XTSFormer as you suggested.
>
> The c-NTPP model is different from our method: c-NTPP utilizes a sequential variational autoencoder framework to identify clusters in latent feature embeddings (in latent feature space), while our XTSFormer framework conducts hierarchical clustering in the time domain in order to capture the complex multi-scale patterns in time domain from real-world data.
>
> We also conducted experiments for comparison on two public datasets. The results are as follows:
>
> |           | Financial |          |      |      | StackOverflow |          |      |      |
> |:---------:|:---------:|:--------:|:----:|:----:|:-------------:|:--------:|:----:|:----:|
> |           | Accuracy (%)| F1-score (%) | RMSE |  NLL |   Accuracy (%) | F1-score (%) | RMSE |  NLL |
> |  c-NTPP [1]|    62.6   |   63.9   | 1.17 | 1.25 |      43.8     |   33.7   | 8.50 | 2.55 |
> | XTSformer |    77.5   |   78.8   | 0.88 | 1.18 |      49.4     |   35.0   | 4.21 | 1.89 |
>
> From the results, we can see that XTSFormer performed better in accuracy over c-NTPP on these two datasets (e.g., with a ~15% increase in accuracy on event type prediction on Financial dataset). This may be due to the importance of capturing multi-scale clustering patterns in the time domain.
>
> [1] Ding, Fangyu, Junchi Yan, and Haiyang Wang. "c-NTPP: learning cluster-aware neural temporal point process." Proceedings of the AAAI Conference on Artificial Intelligence. Vol. 37. No. 6. 2023.
>
>
> ## 2. Comparison with Other Grouping-based Transformers (Q2, W3)
> Thank you for the suggestion. We added comparisons of our XTSFormer model with the efficient transformer models based on grouping tokens into clusters, i.e., Fast Transformers with Clustered Attention [2] and ClusterFormer [3].
>
> After reviewing these two papers in detail, we found that these two works are similar to ours in the sense that they also conduct efficient attention based on clustering. The main difference is that both works conduct clustering in the latent token embeddings (latent feature space), while our method conducts hierarchical clustering in the time domain (to capture the multiple temporal scale patterns). We added experimental comparisons, as you suggested, on the same two public datasets. Please note that we had to recast these two methods into neural temporal point process framework for irregular time event predictions. We maintained consistent hyper-parameters across all models for a fair comparison (details in Appendix A.2.1). The results are listed below.
>
> We can see that while the models by [2] and [3] have better time efficiency, they fall short in prediction accuracy. The results may be due to the importance of capturing multi-scale clustering in time domain (our approach).
>
> |                   |   Financial  |              |      |      |                 | StackOverflow |              |       |      |                 |
> |:-----------------:|:------------:|:------------:|:----:|:----:|:---------------:|:-------------:|:------------:|:-----:|:----:|:---------------:|
> |                   | Accuracy (%) | F1-score (%) | RMSE |  NLL | Time Cost (min) |  Accuracy (%) | F1-score (%) |  RMSE |  NLL | Time Cost (min) |
> | Fast Transformers [2]|     71.6     |     69.2     | 1.57 | 1.44 |       6.9       |      43.5     |     32.8     |  5.17 | 3.04 |       12.4      |
> |   ClusterFormer [3]  |     65.0     |     51.8     | 1.93 | 1.87 |       5.0       |      40.5     |     27.1     | 10.15 | 5.66 |       9.8       |
> |     XTSformer     |     77.5     |     78.8     | 0.88 | 1.18 |       8.4       |      49.4     |     35.0     |  4.21 | 1.89 |       14.3      |
>
>
> [2] Vyas, Apoorv, Angelos Katharopoulos, and François Fleuret. "Fast transformers with clustered attention." Advances in Neural Information Processing Systems 33 (2020): 21665-21674.
> [3] Wang, Ningning, et al. "Clusterformer: Neural clustering attention for efficient and effective transformer." Proceedings of the 60th Annual Meeting of the Association for Computational Linguistics (Volume 1: Long Papers). 2022.
>
> ## 3. Long-range Dependencies (W1)
>
> Thank you for the point. Can you please further elaborate on your concern about our approach's limiting the performance of complex long-range dependencies?

---

> ### Author Response · Authors · 2023-11-22
> **Reminder of Response to Author's Rebuttal**
>
> Dear Reviewer XqYU,
>
> We have thoroughly reviewed each point you raised and have provided detailed responses to address them. We would be grateful if you could kindly review our responses and let us know if they satisfactorily address your concerns. If so, we hope you might consider revising your score accordingly.
>
> Your feedback is immensely valuable to us, and we look forward to any further suggestions you might have.

---

### Meta-Review · Area_Chair_D9w8 · 2023-12-09

**Metareview:**

The paper proposes a method for predicting the time and type of future events based on irregularly timed event data, introducing the Cross-Temporal-Scale Transformer (XTSFormer). The strength lies in that  the paper addresses an interesting problem of event prediction in irregular sequences. The comprehensive ablation study and experimentation on real-world data contribute to the strength of the study.
However, reviewers raised questions regarding the novelty of the proposed method.

Overall, the paper addresses an important problem and introduces innovative components. However, concerns about clarity, novelty leads to the rejection of this paper.

**Justification For Why Not Higher Score:**

However, concerns about clarity, novelty leads to the rejection of this paper.

**Justification For Why Not Lower Score:**

N/A

---

### Decision · Program_Chairs · 2024-01-16

Reject